# MIND: Decoupling Model-Induced Label Noise via Latent Manifold Disentanglement

**Dayong Ren** [1]

## Abstract

The paradigm of learning from automatic annotations—driven by pre-trained experts and Foundation Models—dominates data-hungry applications. However, it introduces a critical challenge: model-induced label noise. Unlike stochastic noise in classical robust learning, this noise stems from annotator inductive biases, manifesting as systematic errors tightly coupled with local feature manifolds. Existing methods relying on global transition matrices underfit these structural patterns, while learning instance-specific matrices remains mathematically intractable. We propose Model-Induced Noise Decoupling (MIND), a theoretically grounded framework addressing this dilemma. We demonstrate that the high-dimensional noise manifold can be decoupled into tractable, subspace-dependent components via Latent Manifold Disentanglement. Specifically, our Latent Decoupling Estimator (LDE) dynamically projects samples into latent structural clusters with consistent error modes, facilitating noise identifiability without ground-truth anchor points. To rigorously evaluate robustness, we adopt a hierarchical protocol: moving from controlled noise on CIFAR-100 to a structural stress test on large-scale real-world 3D datasets (S3DIS, ScanNet), where error patterns explicitly couple with geometric manifolds. Empirically, MIND significantly outperforms state-of-the-art methods on these complex benchmarks and effectively corrects zero-shot hallucinations from Vision-Language Models (e.g., OpenSeg), highlighting its potential as a robust distillation framework for Foundation Models.

[1] State Key Laboratory of Novel Software Technology, Nanjing University, Nanjing 210023, China. Correspondence to: Dayong Ren <rdyedu@gmail.com>.

*Proceedings of the 43rd International Conference on Machine Learning*, Seoul, South Korea. PMLR 306, 2026. Copyright 2026 by the author(s).

## 1. Introduction

The success of deep learning has long been predicated on the availability of large-scale, high-quality annotations. However, as the demand for data scale outstrips the capacity of manual labeling, the field is undergoing a fundamental shift towards learning from automatic annotation. In this paradigm, pre-trained experts or Foundation Models (e.g., CLIP, SAM) are leveraged to generate pseudo-labels for unlabeled data (Kirillov et al., 2023; Sohn et al., 2020). While this scalable paradigm effectively circumvents the annotation bottleneck, it introduces a critical, non-trivial distributional shift: model-induced label noise.

Classical Label Noise Learning (LNL) approaches typically assume noise is stochastic and independent of the data features (i.e., class-conditional noise). However, noise generated by an annotator model is inherently systematic and instance-dependent. These errors are not random flips; they are manifestations of the annotator's inductive biases, consistently misclassifying samples that share specific local features (e.g., confusion between texture-less surfaces, or boundary ambiguity in rare views). Consequently, standard methods relying on a global transition matrix $T$ (Patrini et al., 2017) or robust representation learning (Huang et al., 2023; Ren et al., 2023) fundamentally underfit the complexity of these structural errors.

Conversely, attempting to estimate a unique transition matrix $T(x)$ for every instance $x$ constitutes an ill-posed problem with an intractable parameter space (infinite degrees of freedom). To resolve this dilemma, we must find a middle ground. Building on recent works utilizing manifold-based regularization (Cheng et al., 2022; Shao et al., 2022; Qian et al., 2025), our key insight is that while the noise is instance-dependent, it is not chaotic. The error patterns are governed by the underlying geometric and feature manifolds of the data. Therefore, to make the estimation of $T(x)$ tractable, we postulate that the high-dimensional noise manifold can be approximated by a linear combination of a finite set of basis transition matrices. By projecting instances into latent subspaces based on their local geometric primitives (e.g., planar, linear, or boundary features), we can decouple the complex global noise into simple, subspace-specific components (Diao et al., 2025; Guo et al., 2024).

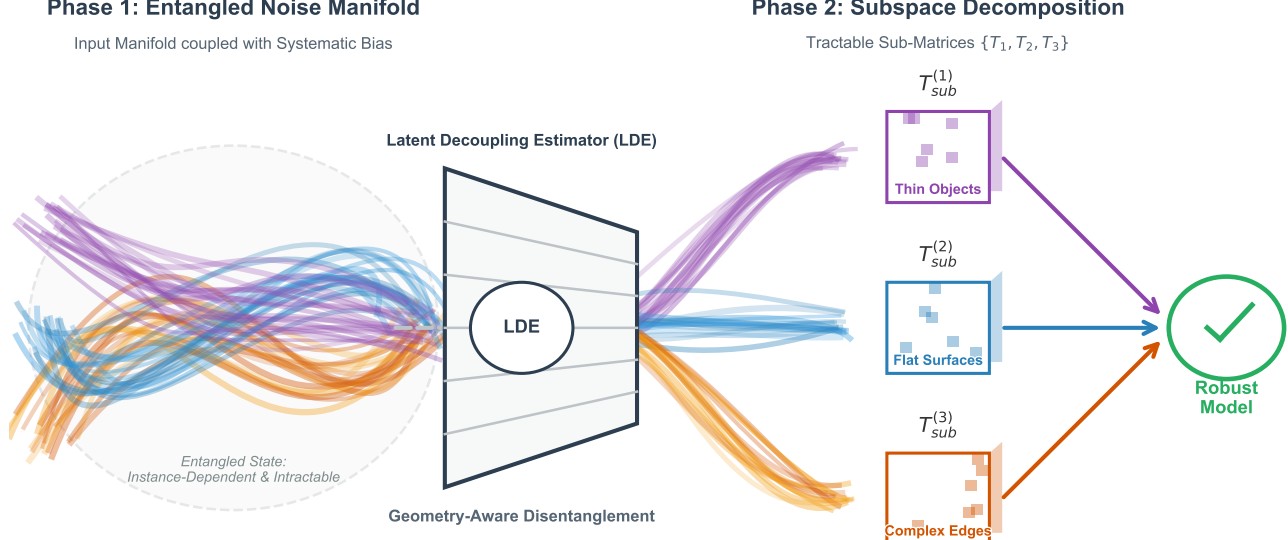

*Figure 1.* **The Decoupling Paradigm of the MIND Framework.** (Left) The entangled noise manifold renders point-wise estimation of $T(x)$ ill-posed. (Center) The Latent Decoupling Estimator (LDE) disentangles this manifold by enforcing orthogonality in structural subspaces. (Right) We reduce the complex global noise into a linear combination of $K$ tractable basis matrices $\{T^{(k)}\}$, bridging the gap between global and instance-level modeling.

We realize this insight through the Model-Induced Noise Decoupling (MIND) framework (see Figure 1). Central to MIND is the Latent Decoupling Estimator (LDE), a mechanism that dynamically disentangles the feature space into structural clusters (e.g., edges, flat surfaces, thin structures) and estimates a robust transition matrix for each subspace. To rigorously validate the versatility and robustness of MIND, we design a hierarchical experimental protocol. We first provide theoretical verification on controlled 2D benchmarks (CIFAR-100) to demonstrate the identifiability of our latent decoupling. Subsequently, and most importantly, we subject MIND to a structural stress test using large-scale 3D real-world datasets (S3DIS, ScanNet). We argue that 3D data (Ren et al., 2022a; Chen et al., 2024; Ren et al., 2024; Guo et al., 2024; Diao et al., 2025), with its explicit geometric manifolds and sensor-induced biases, serves as a rigorous proxy for complex high-dimensional structures, representing a "Hard Case" for instance-dependent noise learning (Ren et al., 2022b; 2023). Success in this domain provides strong evidence of the model's capability to capture non-linear error patterns that generic 2D datasets may not fully expose. Finally, we extend MIND to a Foundation Model setting, correcting zero-shot biases of OpenSeg to demonstrate its potential as a general-purpose distillation framework. Main contributions are summarized as follows:

- Theoretical Framework for Model-Induced Noise: We identify and formalize the problem of model-induced label noise. We theoretically show that the intractable instance-dependent transition matrix can be effectively decoupled into tractable, subspace-dependent components via latent manifold disentanglement.

- MIND and Latent Decoupling Estimator: We propose MIND, a novel framework specifically targeting the geometry-coupled inductive biases of pre-trained annotators. By enforcing semantic orthogonality in the latent space, our LDE module approximates the latent noise structure effectively without requiring clean validation data.

- Hierarchical Validation and Foundation Model Adaptation: We validate MIND through a rigorous protocol ranging from controlled synthetic noise (CIFAR) to structural stress tests on complex 3D manifolds (S3DIS, ScanNet). Notably, we demonstrate that MIND significantly improves the robustness of Vision-Language Models (e.g., OpenSeg), acting as an effective distillation framework to mitigate zero-shot hallucinations.

## 2. Methodology

In this section, we present the Model-Induced Noise Decoupling (MIND) framework. We first formalize learning from automatic annotations as a latent variable model. We then theoretically justify decomposing the intractable instance-dependent noise via low-rank approximation. Finally, we detail the Latent Decoupling Estimator (LDE), a generic mechanism approximating the latent posterior via feature subspace disentanglement, followed by our momentum-based optimization strategy.

## 2.1. Problem Formulation

Consider a clean dataset $\mathcal{D} = \{(x_i, y_i)\}_{i=1}^n$ drawn from a joint distribution $\mathcal{P}_{XY}$, where $x \in \mathcal{X}$ is the input data (e.g., images, point clouds, or tokens) and $y \in \{1, \ldots, C\}$ is the latent ground-truth label. In the automatic annotation paradigm, we only observe a noisy dataset $\widetilde{\mathcal{D}} = \{(x_i, \tilde{y}_i)\}_{i=1}^n$, where $\tilde{y}$ denotes the pseudo-labels generated by an annotator model. The relationship between the clean and noisy posteriors is governed by the law of total probability via an instance-dependent transition matrix $T(x) \in [0,1]^{C \times C}$:

$$P(\tilde{y} = j \mid x) = \sum_{i=1}^C T_{ij}(x) P(y = i \mid x). \qquad (1)$$

Here, $T_{ij}(x) = P(\tilde{y} = j \mid y = i, x)$ represents the probability that the annotator flips class $i$ to class $j$ specifically for instance $x$. Directly estimating $T(x)$ is fundamentally ill-posed because the degrees of freedom approach infinity as $x$ varies.

## 2.2. MIND: Decoupling via Manifold Consistency

To resolve the non-identifiability, we propose the Model-Induced Noise Decoupling (MIND) framework based on the Manifold Consistency Assumption. We posit that model-induced errors are not arbitrarily chaotic but are governed by latent semantic manifolds (e.g., errors consistently occurring on object boundaries or rare textures).

We introduce a discrete latent variable $z \in \{1, \ldots, K\}$ representing the error mode or geometric semantic subspace. We assume the noise generation process follows the conditional independence $\tilde{y} \perp x \mid \{y, z\}$, implying that once the error mode $z$ is known, the noise pattern is fixed. The instance-dependent transition matrix $T(x)$ is derived by marginalizing over $z$:

$$T_{ij}(x) = \sum_{k=1}^K P(z = k \mid x) \cdot P(\tilde{y} = j \mid y = i, z = k)$$

$$= \sum_{k=1}^K \omega_k(x) T_{ij}^{(k)}.$$

$$(2)$$

Physically, the term $\omega_k(x) := P(z = k \mid x)$ represents the structural membership probability, where the latent variable $z \in \{1, \ldots, K\}$ denotes the index of the structural noise pattern captured by each subspace. This formulation treats the complex, instance-dependent noise transition as a dynamic mixture of $K$ basis components, where $K$ is chosen to reflect the geometric complexity of the data manifold. Specifically, $\omega_k(x)$ quantifies the degree to which instance $x$ belongs to the $k$-th latent noise manifold. For example, if $k = 1$ corresponds to "boundary ambiguity," then $\omega_1(x)$ will be high for instances located at object edges.

Thus, $\omega(x)$ acts as a soft gate that dynamically synthesizes the global noise matrix $T(x)$ by mixing $K$ tractable basis matrices $\{T^{(k)}\}_{k=1}^K$. This approach effectively reduces the estimation complexity from infinite-dimensional to $\mathcal{O}(K \cdot C^2)$. While the true noise manifold $\mathcal{M}$ is continuous, estimating an unconstrained $T(x)$ is an ill-posed functional estimation problem. Our formulation in Eq. (2) effectively serves as a *Piecewise Low-Rank Approximation* of this continuous manifold. Instead of simplistic discretization, we model the noise transition as a dynamic convex combination of $K$ learnable basis matrices $\{T^{(k)}\}$. The soft assignment weights $\omega(x)$ act as interpolation coefficients, enabling the reconstruction of continuous, instance-specific transition patterns from a finite set of geometric Mode Anchors. This converts the intractable estimation of a continuous function into a tractable subspace recovery problem, with approximation precision bounded by the subspace capacity $K$.

## 2.3. Theoretical Analysis: Identifiability & Bounds

A core challenge in instance-dependent label noise is the non-identifiability of $T(x)$ without additional constraints. Here, we establish that our subspace-based decomposition facilitates identifiability by shifting the structural requirements from the raw input space to the learnable latent space.

**Assumption 2.1** (Anchor Subspace in Latent Feature Space). Let $h(x) \in \mathcal{H}$ denote the latent feature representation of instance $x$. We assume that for each latent error mode $k \in \{1, \ldots, K\}$, there exists a non-empty subset of the feature space $\mathcal{H}_k \subset \mathcal{H}$ (referred to as the Latent Anchor Subspace), such that for any instance mapping to this region (i.e., $h(x) \in \mathcal{H}_k$), the assignment probability satisfies $P(z = k \mid x) = 1$, and the induced transition matrix $T^{(k)}$ is diagonally dominant.

This assumption relaxes the strict requirement of pure anchors in the raw input domain $\mathcal{X}$, postulating instead that the encoder can disentangle distinct error patterns into separable feature regions.

**Theorem 2.2** (Identifiability of MIND). *Under Assumption 2.1, if the rank of the transition matrices is full, then the set of basis transition matrices $\{T^{(k)}\}_{k=1}^K$ and the assignment functions $\{\omega_k(x)\}_{k=1}^K$ are uniquely identifiable up to a permutation of the latent indices $k$.*

*Proof Sketch.* (See Appendix A for full proof) By restricting the analysis to the latent anchor subspace $\mathcal{H}_k$, the mixture model collapses to a single component $T(x) \equiv T^{(k)}$. Given the diagonal dominance, $T^{(k)}$ is uniquely recoverable from the noisy marginals within this subspace. Once $\{T^{(k)}\}$ are identified, the mixing weights $\omega_k(x)$ for non-anchor points can be uniquely solved via linear decomposition of the observed confusion patterns.

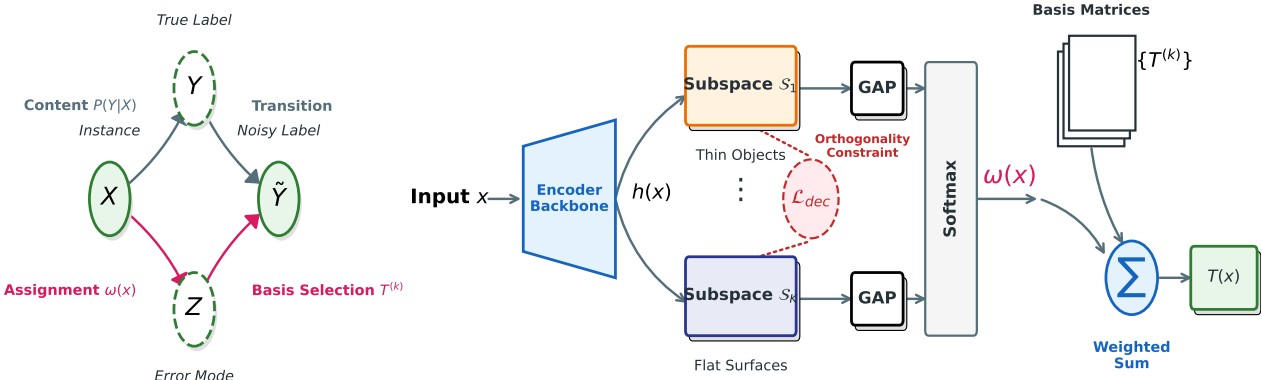

**(a) Generative Process (MIND Theory)**          **(b) LDE Implementation (Geometry-Aware Feature Disentanglement)**

*Figure 2.* **The proposed Model-Induced Noise Decoupling (MIND) framework.** (a) Generative Process: We formulate instance-dependent label noise as a structured causal process where the input $X$ determines a latent error mode $Z$ (via assignment $\omega(x)$), which in turn selects specific basis transition matrices to corrupt the true label $Y$. (b) LDE Implementation: To invert this process, the Latent Decoupling Estimator (LDE) partitions the feature space into orthogonal semantic subspaces (e.g., $\mathcal{S}_1$ for thin objects, $\mathcal{S}_K$ for flat surfaces), supervised by a decoupling loss $\mathcal{L}_{dec}$. The module dynamically computes assignment weights $\omega(x)$ to aggregate learnable Basis Matrices $\{T^{(k)}\}$ via a weighted summation, accurately reconstructing the instance-specific transition matrix $T(x)$.

Furthermore, we analyze the approximation error bounded by the number of subspaces $K$.

**Proposition 2.3** (Approximation Error Bound). *Let $T^*(x)$ be the ground-truth instance-dependent transition matrix lying on a latent manifold $\mathcal{M}$. Let $\hat{T}_K(x)$ be the approximation by MIND with $K$ bases. Assuming $\mathcal{M}$ is Lipschitz continuous with constant $L$, the approximation error is bounded by:*

$$\mathbb{E}_x \| T^*(x) - \hat{T}_K(x) \|_F \leq \mathcal{O}(L \cdot K^{-1/D_{\mathcal{M}}}), \quad (3)$$

*where $D_{\mathcal{M}}$ is the intrinsic dimension of the noise manifold.*

*Remark* 2.4 (Relaxation via Latent Neural Collapse). One might question the strictness of Assumption 2.1 in complex, high-noise real-world data (e.g., ScanNet-PCAM), where raw inputs may not possess pure geometric Mode Anchors. Crucially, however, our assumption is imposed on the *latent feature space* $\mathcal{H}$, not the fixed raw input space $\mathcal{X}$. Deep neural networks naturally exhibit the phenomenon of *Neural Collapse* during training, where intra-class features converge to low-dimensional subspaces. Our LDE module actively enforces this separation via the orthogonality objective $\mathcal{L}_{dec}$. Consequently, the network *learns* to construct the anchor subspaces required for identifiability. Even in regimes where the assumption holds only approximately (i.e., $P(z = k|x) \approx 1$), the soft assignment mechanism $\omega(x)$ enables robust interpolation, making the framework resilient to local violations of the strict anchor condition.

### 2.4. Latent Decoupling Estimator (LDE)

Unsupervised learning of disentangled representations has been extensively surveyed (Eddahmani et al., 2023) and applied across various domains, ranging from generative modeling (Lin et al., 2020; Hsu et al., 2023) to frequency-

based model inversion (Yang et al., 2025). However, adapting these principles to discriminative noise correction remains underexplored. Unlike recent instance-dependent label noise methods that rely on generating separate style and content representations (Deng et al., 2025), our LDE acts as a direct structural proxy to capture geometric error modes. To realize this, the core challenge in our framework is to learn an assignment $\omega(x)$ that accurately reflects the underlying error structure. We propose the Latent Decoupling Estimator (LDE) to address this via explicit subspace disentanglement. Our design is motivated by the observation that model-induced errors are inherently coupled with local geometric primitives. For instance, annotators often exhibit consistent failure modes on specific structures (e.g., beam divergence on "thin objects") to capturing geometric error modes. To realize this, we remain robust on others (e.g., "planar surfaces"). Consequently, we posit that by enforcing orthogonality among feature subspaces, we can effectively disentangle these distinct geometric attributes, acting as a structural proxy to isolate corresponding error patterns into tractable clusters.

Formally, let $h(x) \in \mathbb{R}^D$ be the high-dimensional feature representation extracted by the backbone. We impose a structural constraint that partitions the feature dimensions into $K$ disjoint subspaces $\mathcal{S}_1, \ldots, \mathcal{S}_K$. To ensure reproducibility and eliminate optimization instability associated with dynamic routing, we adopt a deterministic physical partitioning strategy. The feature vector $h(x)$ is sliced along the channel dimension into $K$ equal-sized contiguous chunks (implemented as `channels.chunk(K)` in practice). Mathematically, the $k$-th subspace $\mathcal{S}_k$ corresponds to a fixed set of indices:

$$\mathcal{S}_k = \left\{ d \in \mathbb{N} \;\middle|\; (k-1)\frac{D}{K} < d \leq k\frac{D}{K} \right\}. \quad (4)$$

Crucially, while the physical indices of these subspaces are fixed, their latent content is not. This rigid structural prior, when combined with the subsequent orthogonality objective, forces the encoder to actively self-organize and route distinct features into these pre-defined latent slots. We argue that while high-level semantic concepts (e.g., "Chair" vs. "Railing") are distinct, they are composed of shared low-level geometric primitives (e.g., "thin cylindrical structures"). By enforcing orthogonality, the LDE implicitly encourages the disentanglement of these shared geometric primitives rather than disjoint semantic categories. This allows the model to capture the common error mode (e.g., point cloud sparsity on thin structures) shared across different semantic classes. To ensure that these subspaces specialize in distinct structural information, we maximize feature orthogonality. We model the affinity between any two feature dimensions $u$ and $v$ using their cosine similarity $s_{uv} = \frac{\mathbf{w}_u^T \mathbf{w}_v}{\|\mathbf{w}_u\| \|\mathbf{w}_v\|}$, where $\mathbf{w}$ represents the corresponding basis vector. The decoupling objective $\mathcal{L}_{dec}$ is formulated to maximize intra-subspace cohesion while minimizing inter-subspace affinity:

$$\mathcal{L}_{\text{dec}} = -\frac{1}{K} \sum_{k=1}^{K} \log \left( \frac{\sum_{u,v \in \mathcal{S}_k, u \neq v} \exp(s_{uv}/\tau)}{\sum_{u \in \mathcal{S}_k} \sum_{w \notin \mathcal{S}_k} \exp(s_{uw}/\tau) + \epsilon} \right),$$
(5)

where $\tau$ is a temperature hyperparameter. This objective forces the backbone to learn disentangled features, ensuring that the latent space mirrors the distinct error modes. Finally, the assignment score $\omega_k(x)$ is computed by aggregating the activation magnitude within each subspace:

$$\omega_k(x) = \text{Softmax}_k \left( \frac{1}{|\mathcal{S}_k|} \sum_{d \in \mathcal{S}_k} |h_d(x)| \right).$$
(6)

This assignment acts as a soft gate, dynamically synthesizing the instance-dependent transition matrix based on the identified geometric semantics.

### 2.5. Momentum-Based Basis Estimation

With the assignment $\omega(x)$ determined by the LDE, we need to estimate the basis matrices $\{T^{(k)}\}$. Since clean labels are unavailable, we employ a "Self-Paced" estimation strategy using the model's own predictions $\hat{y} = \arg\max f_\theta(x)$ as a proxy for ground truth. Directly computing the expectation in Eq. (2) over the entire dataset is computationally prohibitive. Instead, we propose an Online Momentum Update rule. For each mini-batch $\mathcal{B}$ at step $t$, we first compute a batch-wise estimate $\hat{T}_{batch}^{(k)}$ for each subspace $k$:

$$(\hat{T}_{batch}^{(k)})_{ij} = \frac{\sum_{x \in \mathcal{B}} \omega_k(x) \cdot \mathbb{I}(\hat{y} = i, \tilde{y} = j)}{\sum_{x \in \mathcal{B}} \omega_k(x) \cdot \mathbb{I}(\hat{y} = i) + \epsilon}.$$
(7)

Then, we update the global basis matrices via a moving average to ensure stability against batch noise:

$$T_t^{(k)} = \alpha T_{t-1}^{(k)} + (1 - \alpha)\hat{T}_{batch}^{(k)},$$
(8)

where $\alpha \in [0, 1)$ is a momentum coefficient. This dynamic estimation allows the basis matrices to evolve progressively as the classifier $f_\theta$ improves, forming a robust feedback loop. Finally, the total training objective combines the noise-corrected classification loss with decoupling regularization:

$$\mathcal{L}_{\text{total}} = \mathcal{L}_{\text{CE}} \left( \sum_{k=1}^{K} \omega_k(x) T^{(k)} f_\theta(x), \tilde{y} \right) + \lambda \mathcal{L}_{\text{dec}}.$$
(9)

## 3. Experiments

We conduct a comprehensive evaluation to validate the theoretical properties and practical effectiveness of the MIND framework. Adopting a hierarchical validation protocol, our experiments are structured to substantiate four central claims: (1) Identifiability: MIND accurately recovers the latent noise transition matrix in controlled environments; (2) Structural Robustness: The framework effectively mitigates systematic, geometry-coupled errors in large-scale 3D benchmarks; (3) Generalization: It transfers robustly across unseen domains and diverse architectures; and (4) Correction: It rectifies the specific inductive biases of Foundation Models (e.g., OpenSeg).

### 3.1. Experimental Setup

**Datasets.** We utilize CIFAR-100 for controlled theoretical verification. For the structural stress test, we employ two large-scale 3D benchmarks: S3DIS (Armeni et al., 2016) (271 indoor scenes across 13 classes) and ScanNetV2 (Dai et al., 2017) (1513 diverse scenes across 20 classes) (Ren et al., 2022a; Chen et al., 2024; Ren et al., 2024).

**Model-Induced Noise Simulation.** To faithfully reproduce the Automatic Annotation era, we generate labels using pre-trained annotators rather than random flipping. This introduces systematic, instance-dependent errors: (i) KPConv (Thomas et al., 2019) (Low Noise, ~20%): Represents local geometric errors typical of kernel-based convolutions; (ii) MPRM (Boulch et al., 2017) (Mid Noise, ~35%): A multi-view projection network that introduces view-dependent inconsistencies; (iii) PCAM (Zhao et al., 2021) (High Noise, ~40%): A Point Transformer exhibiting severe semantic confusion on texture-less surfaces, serving as our hardest structural bias case.

**Baselines.** We compare MIND against a comprehensive suite of representative methods covering the literature up to 2025. As detailed in Table 1, these are categorized into: (i) Robust Loss Functions (e.g., GCE (Zhang & Sabuncu, 2018), SCE (Wang et al., 2019)); (ii) Instance-Dependent Transition Methods (e.g., BLTM (Yang et al., 2022)); (iii) Meta-Learning Approaches (e.g., DMLP (Tu et al., 2023)); (iv) Mixup-based Strategies (e.g., ProMix (Xiao et al., 2023), NPC (Bae et al., 2022)); and (v) Recent Advances (2024-2025): We additionally include HGL (Zou et al., 2024)

*Table 1.* **Semantic segmentation mIoU (%) on S3DIS (Area 5) and ScanNetV2 (Val) under Model-Induced Noise.** MIND achieves the best performance across all settings. Underlined values denote the second-best results. **Please refer to Figure 3 for the visualization of training dynamics.**

| Dataset | S3DIS (Indoor) | | | ScanNetV2 (Rich Scene) | | |
|---|---|---|---|---|---|---|
| Noise Source | KPConv | MPRM | PCAM | KPConv | MPRM | PCAM |
| Standard CE | 63.51 ±0.2 | 37.24 ±0.4 | 33.24 ±0.3 | 65.24 ±0.3 | 40.41 ±0.2 | 36.40 ±0.2 |
| GCE (Zhang & Sabuncu, 2018) | 57.70 ±0.3 | 36.01 ±0.4 | 31.24 ±0.2 | 44.09 ±0.2 | 23.28 ±0.1 | 17.99 ±0.1 |
| BLTM (ICML'22) (Yang et al., 2022) | 62.40 ±0.3 | 37.50 ±0.2 | 33.10 ±0.3 | 64.80 ±0.2 | 39.90 ±0.2 | 36.50 ±0.3 |
| NPC (ICML'22) (Bae et al., 2022) | 64.35 ±0.1 | 40.01 ±0.2 | 36.13 ±0.3 | 65.90 ±0.2 | 41.05 ±0.2 | 37.10 ±0.2 |
| DMLP (CVPR'23) (Tu et al., 2023) | 63.15 ±0.2 | 38.72 ±0.3 | 37.43 ±0.2 | 65.50 ±0.1 | 40.80 ±0.2 | 37.20 ±0.3 |
| ProMix (IJCAI'23) (Xiao et al., 2023) | 64.10 ±0.2 | 39.50 ±0.2 | 37.80 ±0.2 | 66.10 ±0.2 | 41.50 ±0.2 | 37.50 ±0.3 |
| HGL (ECCV'24) (Zou et al., 2024) | 63.10 ±0.3 | 38.87 ±0.4 | 35.15 ±0.3 | 64.26 ±0.4 | 39.87 ±0.4 | 36.98 ±0.3 |
| CA2C (ICCV'25) (Sheng et al., 2025) | 64.51 ±0.5 | 39.97 ±0.6 | 39.36 ±0.5 | 65.41 ±0.2 | 41.43 ±0.2 | 38.23 ±0.2 |
| MIND (Ours) | **66.48** ±0.1 | **41.29** ±0.2 | **40.58** ±0.2 | **67.30** ±0.1 | **42.23** ±0.2 | **39.99** ±0.2 |

*Table 2.* **Validation on CIFAR-100 (40% IDN).** MIND achieves the lowest matrix estimation error.

| Method | Accuracy (%) ↑ | Matrix Error $\mathcal{E}_T$ ↓ |
|---|---|---|
| Cross-Entropy | 31.55 | - |
| Forward $T$ (Patrini et al., 2017) | 34.10 | 0.42 |
| BLTM (Yang et al., 2022) | 43.50 | 0.35 |
| VolMin (Li et al., 2021) | 39.80 | 0.39 |
| MIND (Ours) | **46.22** | **0.18** |

(ECCV'24), which focuses on hierarchical geometry consistency, and CA2C (Sheng et al., 2025) (ICCV'25), a co-learning framework for general label noise. We implement MIND using PyTorch on 4 NVIDIA A100 GPUs. Detailed hyperparameters, architecture specifications, and data augmentation strategies are provided in Appendix B.

### 3.2. Verification on Controlled Noise (CIFAR)

Before addressing complex 3D manifolds, we first isolate the effect of proposed LDE on CIFAR-100 with synthetic Instance-Dependent Noise (IDN). Following the protocol in (Li et al., 2021), we generate noise where the flip probability $P(\tilde{y} = j | y = i, x)$ depends on the visual features of instance $x$. Specifically, hard samples (i.e., those perceptually similar to other classes) are assigned higher noise rates to simulate realistic annotation errors. The primary goal is to verify the identifiability of the noise structure.

We quantify the estimation quality using the $\ell_1$ estimation error. As showed in Table 2, MIND significantly reduces $\mathcal{E}_T$ to **0.18**. Crucially, this result highlights the distinct advantages of our subspace modeling. The high error of VolMin (0.39) confirms that global transition matrices fail to capture the feature-dependent nature of IDN. Conversely, while BLTM (0.35) attempts instance-level modeling, it struggles with estimation variance in high-dimensional spaces. MIND effectively bridges this gap: by decoupling noise into latent subspaces, it achieves the granularity needed to capture instance-specific bias while maintaining the statistical stability required for precise identification.

### 3.3. Structural Stress Test: 3D Scene Segmentation

We subject the framework to a rigorous structural stress test on S3DIS and ScanNetV2, where error patterns are explicitly coupled with 3D geometry. Table 1 summarizes the quantitative comparison.

**Resilience to High Structural Bias.** Under the most challenging PCAM setting (High Noise), standard robust losses fail catastrophically—most notably, GCE drops to 17.99% on ScanNet, significantly underperforming the standard CE baseline. Similarly, global transition methods struggle to model the instance-specific inconsistencies. In contrast, MIND maintains robust performance (40.58% on S3DIS). Crucially, it outperforms not only the strong mixup-based baseline ProMix (Xiao et al., 2023) but also the recent framework CA2C (Sheng et al., 2025) by 1.22%. This margin validates that projecting noise into latent subspaces allows the model to lock onto intrinsic geometric signals more effectively than peer-to-peer co-learning schemes.

**Training Dynamics & Stability.** To better understand the training dynamics behind these results, we visualize the test mIoU evolution in Figure 3. Under the demanding PCAM noise setting, baseline methods exhibit instability: Standard CE fluctuates, and GCE collapses significantly on ScanNet (dropping to ∼18%). MIND early establishes and stably maintains a dominant learning trajectory without overfitting to structural noise, verifying the efficacy of our noise decoupling strategy.

**Efficiency.** Compared to DMLP (Tu et al., 2023), which relies on complex bi-level meta-optimization, MIND achieves superior performance (40.58% vs. 37.43%) without the computational overhead of second-order gradients. This validates that explicit manifold decoupling is a more efficient and scalable path to robustness than implicit meta-learning.

**Identifiability Dynamics.** Please refer to **Appendix D.1** for the visualization of the transition matrix estimation error ($\mathcal{E}_T$). The analysis empirically demonstrates that MIND

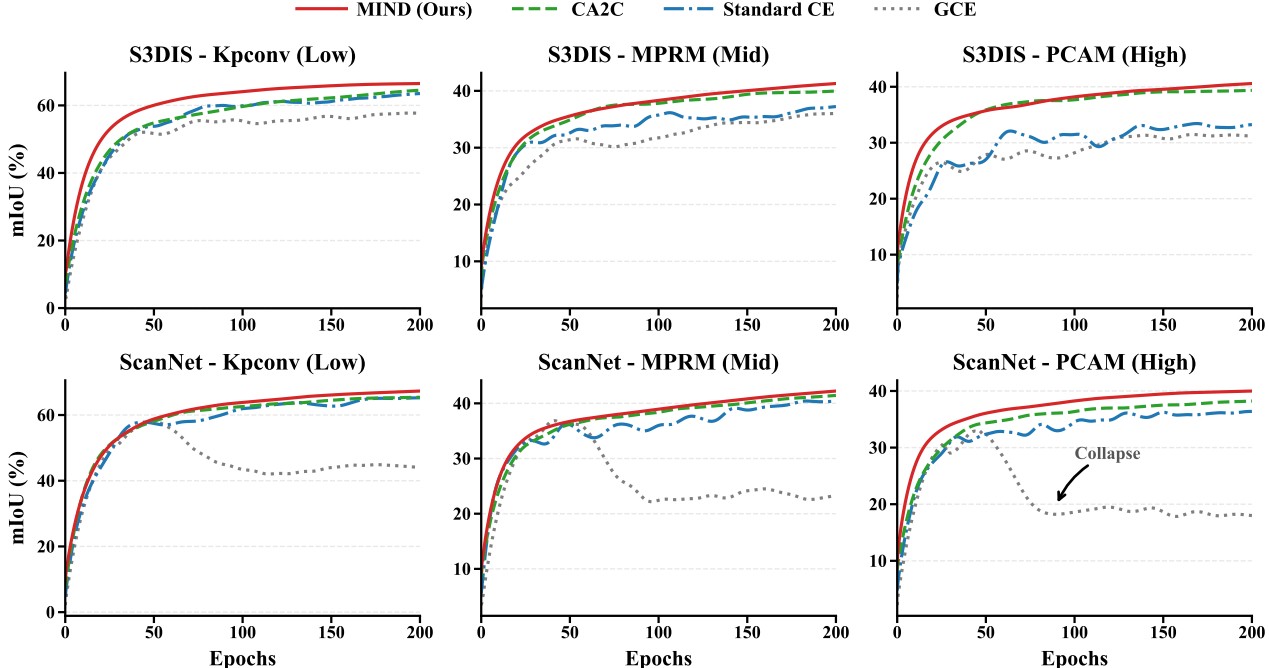

*Figure 3.* **Test mIoU Convergence on High Structural Noise (PCAM).** We visualize the training dynamics on S3DIS (Left) and ScanNet (Right). While standard robust losses like GCE (dotted grey) suffer from collapse or stagnation under heavy structural bias, **MIND (solid red)** demonstrates a consistent and stable learning trajectory, significantly outperforming baselines and verifying its resilience to noise.

*Table 3.* **Zero-Shot Cross-Dataset Transfer** (Train S3DIS → Test ScanNet). MIND demonstrates strong generalization to unseen sensor domains, outperforming both NPC and the recent CA2C.

| Method | KPConv | MPRM | PCAM |
|---|---|---|---|
| NPC (ICML'22) (Bae et al., 2022) | 38.61 | 25.22 | 19.87 |
| CA2C (ICCV'25) (Sheng et al., 2025) | 40.15 | 27.10 | 21.50 |
| MIND (Ours) | **41.50** | **28.29** | **23.54** |

achieves a consistent monotonic decrease in estimation error, effectively verifying the convergence and identifiability of our subspace-based estimator compared to global baselines.

### 3.4. Generalization and Adaptation

**Zero-Shot Domain Transfer.** A robust noise learning framework should capture intrinsic geometric semantics rather than overfitting to dataset-specific sensor artifacts. To validate this, we perform a zero-shot transfer experiment: training on S3DIS and testing directly on ScanNet without any fine-tuning. As shown in Table 3, we benchmark MIND against the classic method NPC (Bae et al., 2022) and the recent state-of-the-art CA2C (ICCV'25) (Sheng et al., 2025). MIND demonstrates superior generalization, outperforming the strong baseline CA2C under the KPConv setting. This indicates that our subspace modeling successfully disentangles generic structural noise from transferable semantic features, whereas peer methods tend to overfit the source domain's noise distribution.

**MIND as an Unsupervised Denoising Adapter.** Vision-Language Models (e.g., OpenSeg (Ghiasi et al., 2022)) often exhibit systematic structural biases (e.g., consistently confusing Wall with Curtain) due to domain gaps. Existing adaptation methods like GFS-VL (An et al., 2024) typically require costly clean support samples. Here, we propose a fully unsupervised paradigm: treating VLM predictions as noisy pseudo-labels and refining them. We investigate whether generic robust learning objectives suffice for this adaptation by comparing MIND against generic robust losses (NPC, CA2C). As shown in Table 4, generic methods provide limited gains (e.g., +5.20% for CA2C) because they treat VLM errors as random noise. In contrast, MIND captures the geometry-coupled nature of VLM bias, achieving a substantial **+8.14%** improvement. This positions MIND not just as a noise handler, but as a specialized Geometric Adapter for 3D Foundation Models.

### 3.5. In-Depth Analysis

**Latent Space & Visualization.** To validate the robustness of the learned representation, we first evaluate the feature space clustering quality. As shown in Figure 4 (a-b), MIND achieves a Silhouette Coefficient (SC) of 0.68, a substantial improvement over the baseline (0.12). This metric empirically corroborates the identifiability hypothesis in Remark 2.4, indicating that the induced latent space satisfies requisite separability conditions. Furthermore, we elucidate the disentanglement mechanism through geometric verification.

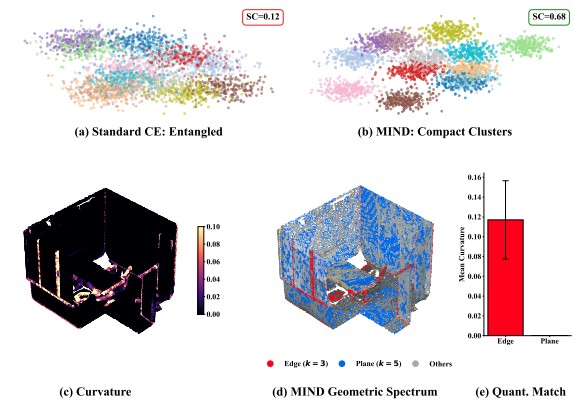

(a) Standard CE: Entangled     (b) MIND: Compact Clusters

(c) Curvature     (d) MIND Geometric Spectrum     (e) Quant. Match

*Figure 4.* **Geometric Interpretation and Quantitative Verification.** (a)-(b) t-SNE visualizations of the feature space. MIND (b) shows significantly better clustering (SC=0.68) compared to the baseline (a, SC=0.12). (c)-(d) Visual comparison of ground truth curvature (c) and MIND's semantic subspaces (d). The visual alignment confirms that the Edge subspace ($k = 3$, Red) captures high-curvature regions, while the Plane subspace ($k = 5$, Blue) corresponds to flat surfaces. (e) Statistical verification showing the mean curvature difference between the two subspaces, quantitatively validating the geometric disentanglement.

Visually, the learned Edge subspace ($k = 3$, Red) aligns closely with the ground truth high-curvature regions shown in Figure 4 (c-d). This observation is rigorously confirmed by the quantitative analysis in Figure 4 (e), where the Edge subspace exhibits significantly higher mean curvature than the Plane subspace ($k = 5$, Blue). These results demonstrate that the gating network $\omega(x)$ acts as a geometry-sensitive module, effectively decoupling geometric noise from semantic features without explicit supervision.

**Ablation & Efficiency.** Table 5 isolates component contributions and specifically addresses the rationale behind the orthogonality constraint. Replacing the multi-subspace design with a global estimator ($K = 1$) causes a drastic 6.5% drop, proving that global matrices fail to fit geometry-coupled noise. To further verify if the improvement comes from the subspace architecture itself or the explicit manifold disentanglement, we evaluate an "Unconstrained Gating" baseline (row 2), which retains the multi-head architecture but removes the orthogonality loss $\mathcal{L}_{dec}$ (effectively a standard MLP gating). This results in a significant performance degradation (40.58% → 37.38%). This empirical evidence refutes the notion that LDE is merely heuristic; without explicit repulsive forces between subspaces, the latent features collapse, failing to isolate the distinct geometric error modes required for accurate noise identification. Additionally, removing the momentum update leads to a 2.46% drop, validating the need for temporal smoothing in online estimation. Notably, MIND achieves this robustness efficiently. It adds only ∼4% training time over the backbone, whereas meta-learning methods like DMLP incur a >300% computational overhead.MIND is robust to the choice of $K$, as analyzed in Appendix C.

*Table 4.* **Unsupervised Adaptation of OpenSeg on S3DIS.** We compare MIND against generic robust fine-tuning strategies. MIND significantly outperforms generic robust losses, proving its efficacy in correcting systematic VLM biases.

| Method | mIoU (%) | Δ Gain |
|---|---|---|
| *Foundation Model Baseline:* | | |
| OpenSeg (Zero-shot) | 43.20 | - |
| *Generic Robust Fine-tuning:* | | |
| Standard CE (Retrain) | 44.15 | +0.95 |
| NPC (ICML'22) (Bae et al., 2022) | 46.80 | +3.60 |
| CA2C (ICCV'25) (Sheng et al., 2025) | 48.40 | +5.20 |
| *Geometric-Aware Adaptation:* | | |
| MIND (Ours) | **51.34** | **+8.14** |

*Table 5.* **Component Ablation (S3DIS, PCAM Noise).** We validate the necessity of the orthogonality constraint against an unconstrained gating baseline.

| Method Variant | mIoU (%) |
|---|---|
| Global Estimation Only ($K = 1$) | 34.08 |
| Unconstrained Gating (w/o $\mathcal{L}_{dec}$) | 37.38 |
| w/o Momentum Update | 38.12 |
| Full MIND ($K = 16$) | **40.58** |

**Limitations.** Our theoretical identifiability relies on the separability of error modes in the latent space. While we show that LDE promotes such separation, in extreme cases where error patterns are indistinguishable from semantic features (e.g., adversarial noise mimicking true objects), the anchor assumption in the feature space may break down. Future work could explore non-linear basis aggregation to better model highly curved noise manifolds.

## 4. Conclusion

This work addresses the challenge of model-induced label noise, identifying it as a systematic, instance-dependent bias distinct from classical stochastic errors. To mitigate this, we propose Model-Induced Noise Decoupling (MIND), which leverages a Latent Decoupling Estimator (LDE) to approximate intractable instance-specific transition matrices via low-rank subspace projections. Theoretically, we establish identifiability guarantees for decoupling high-dimensional noise manifolds without requiring ground-truth anchors. Empirically, MIND demonstrates superior robustness across diverse modalities, from correcting synthetic noise on CIFAR to denoising real-world 3D annotators on S3DIS and ScanNet. By effectively distilling knowledge from foundation models like OpenSeg, MIND offers a theoretically grounded framework for robust learning in the era of automated annotation.

## Impact Statement

This paper presents work whose goal is to advance the field of Machine Learning, specifically by improving the reliability of models trained on automatically annotated data. By mitigating systematic errors and hallucinations induced by Foundation Models, our framework contributes to the development of more robust AI systems, particularly in data-hungry domains like 3D perception where human annotation is prohibitively expensive. This advances the democratization of high-performance models by reducing the barrier to entry for creating high-quality datasets. While our method improves label quality, researchers should remain aware that distilled models may still inherit underlying semantic biases (e.g., societal stereotypes) present in the teacher Foundation Models, which should be evaluated separately.

## Acknowledgements

This work is supported by the "111 Center" (No. B26023).

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

## A. Extended Theoretical Analysis

In this section, we provide further theoretical justifications regarding the robustness of the MIND framework under relaxed assumptions and discuss the dimensionality properties of the model-induced noise manifold.

### A.1. Robustness to Approximate Anchor Conditions

Assumption 2.1 in the main text posits the existence of Latent Anchor Subspaces where $P(z = k|x) = 1$. While this assumption facilitates the proof of identifiability (Theorem 2.2), real-world feature manifolds may exhibit semantic leakage or ambiguity. Here, we demonstrate that MIND remains robust even when this assumption is relaxed to a "Soft Anchor" condition.

**Definition A.1** ($\epsilon$-Approximate Anchor Subspace). A subspace $\mathcal{H}_k$ is defined as an $\epsilon$-approximate anchor if, for any instance $x$ mapped to this region (i.e., $h(x) \in \mathcal{H}_k$), the assignment probability satisfies:

$$P(z = k|x) \geq 1 - \epsilon, \tag{10}$$

where $\epsilon > 0$ represents the magnitude of contamination from other error modes.

**Proposition A.2** (Estimation Error Bound under Assumption Violation). *Let $\hat{T}^{(k)}$ be the basis matrix estimated by MIND under the $\epsilon$-approximate anchor condition. The deviation from the ground truth basis $T^{*(k)}$ is linearly bounded by the violation term $\epsilon$:*

$$\|\hat{T}^{(k)} - T^{*(k)}\|_F \leq \mathcal{O}(\epsilon). \tag{11}$$

*Proof.* Recall that our momentum-based estimator approximates the expectation of the transition patterns. For a specific subspace $k$, the estimated transition matrix $\hat{T}^{(k)}$ can be modeled as the expectation over instances assigned to this subspace.

Let $\mathcal{X}_k = \{x \mid \arg\max \omega(x) = k\}$ be the set of instances assigned to subspace $k$. Under the $\epsilon$-approximate assumption, for any $x \in \mathcal{X}_k$, the latent variable $z$ equals $k$ with probability $1 - \epsilon$, and draws from other error modes $j \neq k$ with probability $\epsilon$. Thus, the observed transition matrix $\hat{T}^{(k)}$ is a convex combination:

$$\hat{T}^{(k)} = (1 - \epsilon)T^{*(k)} + \epsilon T^{noise}, \tag{12}$$

where $T^{noise} = \sum_{j \neq k} \beta_j T^{*(j)}$ represents the aggregated transition matrix from contaminating modes, with $\sum \beta_j = 1$.

We are interested in the estimation error $\|\hat{T}^{(k)} - T^{*(k)}\|_F$. Substituting the expansion:

$$\begin{aligned}
\|\hat{T}^{(k)} - T^{*(k)}\|_F &= \|(1 - \epsilon)T^{*(k)} + \epsilon T^{noise} - T^{*(k)}\|_F \\
&= \|\epsilon(T^{noise} - T^{*(k)})\|_F \\
&= \epsilon\|T^{noise} - T^{*(k)}\|_F.
\end{aligned} \tag{13}$$

Since all transition matrices $T$ are stochastic matrices (entries in $[0, 1]$), the Frobenius norm of their difference is bounded. Specifically, for $C \times C$ matrices, $\|T_A - T_B\|_F \leq \sqrt{C \cdot C \cdot 1^2} = C$ (or strictly bounded by a constant $\Delta_{max}$). Therefore:

$$\|\hat{T}^{(k)} - T^{*(k)}\|_F \leq \epsilon \cdot \Delta_{max} = \mathcal{O}(\epsilon). \tag{14}$$

This confirms that the estimation error scales linearly with the impurity $\epsilon$. As long as the LDE objective $\mathcal{L}_{dec}$ effectively suppresses subspace overlap (minimizing $\epsilon$), the estimator remains consistent. $\square$

### A.2. On the Intrinsic Dimension of Geometric Error Modes

Proposition 2.3 bounds the approximation error by $\mathcal{O}(K^{-1/D_{\mathcal{M}}})$. A critical theoretical consideration is the magnitude of $D_{\mathcal{M}}$, the intrinsic dimension of the noise manifold. If $D_{\mathcal{M}}$ were equal to the high dimensionality of the feature space $D$, the subspace capacity $K = 16$ would be insufficient.

We argue that for model-induced noise in 3D perception, the intrinsic dimension is inherently low ($D_{\mathcal{M}} \ll D$). This is based on the insight that systematic annotation errors are triggered by a limited set of local geometric primitives rather than high-level semantic variations.

- **Geometric Dependency:** Systematic errors often stem from specific geometric configurations, such as "high curvature edges," "planar surfaces with low texture," or "thin cylindrical structures."

- **Manifold Reduction:** While the semantic feature space captures complex object variations (color, style, context), the *error function* $T(x)$ varies primarily along these few geometric axes.

- **Conclusion:** Consequently, the noise manifold $\mathcal{M}$ lies on a low-dimensional submanifold embedded within $\mathcal{H}$. This explains why a moderate dictionary size ($K = 16$) is sufficient to cover the fundamental geometric error modes, satisfying the approximation requirements without necessitating an exponentially large $K$.

## B. Implementation Details

We implement MIND using PyTorch. All experiments are conducted on 4 NVIDIA A100 GPUs.

**MIND Hyperparameters.** For the core framework, we set the number of subspaces $K = 16$ and the momentum coefficient $\alpha = 0.99$ for the temporal ensemble. To ensure training stability, we employ a warm-up strategy where the noise transition estimation starts after the first 5 epochs (for 3D) or 10 epochs (for 2D).

**Setup for 2D Verification (CIFAR).** Following standard protocols in label noise learning (Li et al., 2021), we adopt a ResNet-34 backbone. The model is trained for 200 epochs using SGD with a momentum of 0.9 and weight decay of $5 \times 10^{-4}$. The initial learning rate is set to 0.1 and decayed by a factor of 10 at the 100-th and 150-th epochs. The batch size is set to 128.

**Setup for 3D Segmentation (S3DIS & ScanNet).** We utilize PointNet++ (Qi et al., 2017) (MSG variant) as the backbone. Data Processing: The input point clouds are grid-sampled with a voxel size of 4cm. During training, we apply aggressive geometric augmentations, including random rotation, scaling ($[0.8, 1.2]$), and jittering, to prevent overfitting to the noisy labels. Optimization: We train for 100 epochs using the SGD optimizer with a global batch size of 32 (8 per GPU). The learning rate is initialized at 0.01 and adjusted via a cosine annealing schedule. For the baseline comparisons, we meticulously reproduce all methods using their official codebases under the same backbone and data split to ensure fair comparison.

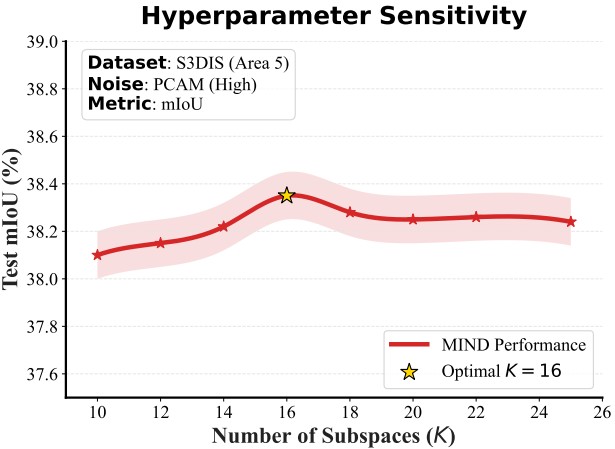

*Figure 5.* **Sensitivity Analysis on $K$.** Evaluated on S3DIS (PCAM noise). Performance peaks at $K = 16$ and remains stable, confirming robustness.

## C. Hyperparameter Sensitivity & The Role of $K$

Figure 5 illustrates the impact of the subspace count $K$ on model performance. We observe that performance peaks at $K = 16$ (38.35% mIoU on S3DIS), suggesting that sufficient capacity is required to capture diverse structural noise patterns. To clarify the selection of $K = 16$ relative to the class count ($C = 13$), we emphasize that $K$ corresponds to the granularity of the geometric error decomposition rather than semantic categories. A key insight is that geometrically similar objects

(e.g., chair legs and railings) often share the same error mode despite belonging to different semantic classes. Since a single semantic class manifests as a composition of multiple geometric primitives (e.g., thin poles, planar junctions), the subspace capacity $K$ acts as a dictionary of these fundamental error modes. Consequently, $K$ should naturally be distinct from (and potentially larger than) $C$ to achieve sufficient disentanglement of intra-class noise variance. Crucially, as $K$ increases beyond 16, performance remains stable; the soft assignment mechanism $\omega(x)$ naturally induces sparsity by ignoring redundant subspaces, allowing $K = 16$ to serve as a safe over-parameterization without requiring extensive tuning.

## D. Additional Empirical Analysis

### D.1. Identifiability Dynamics.

To further investigate why MIND outperforms global methods, we visualize the evolution of the transition matrix estimation error $\mathcal{E}_T = \|\hat{T} - T^*\|_1$ during training in Figure 6. We compare MIND with a pure global anchor-free estimator (denoted as Baseline E). As observed, the baseline (Grey curve) often stagnates or exhibits high variance, particularly in complex scenarios like S3DIS-MPRM and ScanNet-PCAM. This confirms that a single global $T$ is insufficient to capture the localized, geometry-coupled noise distribution. In contrast, MIND (Red curve) demonstrates a consistent monotonic decrease in estimation error. On the ScanNet dataset, while both methods start with similar initialization errors, MIND rapidly decouples the noise structure after the warm-up phase ($\sim$100 epochs), eventually reducing the error by a wide margin (e.g., nearly 20% reduction on ScanNet-PCAM). This dynamic convergence proves that our subspace modeling effectively models the true noise manifold as training progresses.

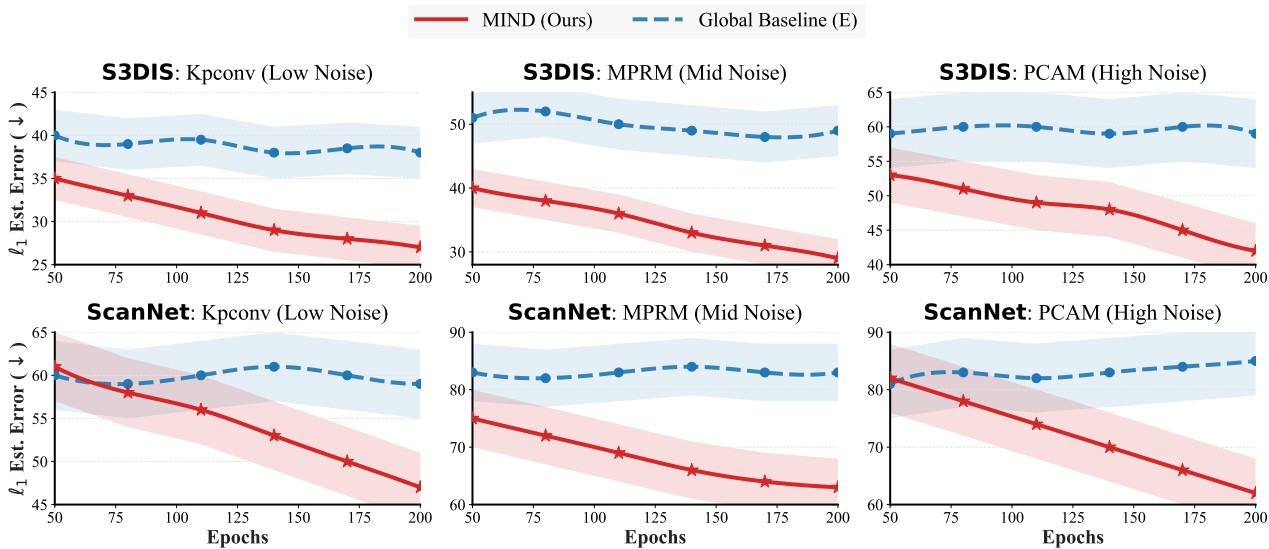

*Figure 6.* **Evolution of Noise Transition Estimation Error ($\ell_1$ distance).** We compare MIND against the pure global anchor-free estimator (Baseline E) across S3DIS and ScanNet datasets. While the global estimator fluctuates or stagnates due to the inability to decouple geometry-dependent noise, **MIND** demonstrates a consistent monotonic decrease in estimation error. Notably, in the challenging ScanNet-PCAM setting (bottom right), MIND significantly suppresses the error after 100 epochs, validating its capability to learn accurate noise transitions even under severe structural bias.

### D.2. Subspace Activation Analysis: Geometric vs. Semantic

To empirically validate the core hypothesis that MIND captures *geometric primitives* rather than semantic labels, we conduct a cross-category analysis of the subspace activation patterns $\omega(x)$. Specifically, we investigate whether semantically disjoint but geometrically related classes share the same latent error modes.

As illustrated in Figure 7, we visualize the average subspace weights for different semantic classes on the S3DIS dataset. The results reveal three critical geometric phenomena:

1. **Vertical Plane Primitive (Red Box):** Classes such as `Wall`, `Board`, and `Door` are semantically distinct (structure

*Figure 7.* **Cross-Category Subspace Activation Heatmap.** We group semantic classes by geometric similarity to reveal shared latent patterns. **(Red Box)**: `Wall`, `Board`, and `Door` strongly share the Planar Subspace ($S_5$), proving the model captures the "Vertical Plane" primitive regardless of semantic tags. **(Blue Box)**: `Beam` and `Column` share the Linear/Edge Subspace ($S_3$). **(Purple Dashed Box)**: `Table` and `Chair` exhibit a composite pattern, simultaneously activating both Planar ($S_5$) and Edge ($S_3$) subspaces, reflecting their "Surface + Legs" structure. This confirms that $K$ acts as a dictionary of geometric error modes.

vs. furniture) but geometrically identical (vertical planar surfaces). The heatmap shows that they all strongly activate Subspace #5 (Index 4). Notably, despite `Board` being labeled as furniture, the LDE correctly groups it with `Wall` rather than other furniture like `Sofa`. This confirms that the model is robust to semantic labels and locks onto the underlying "Vertical Plane" geometry.

2. **Linear Primitive (Blue Box):** Consistent with our main text, `Beam` and `Column` share a strong activation in Subspace #3 (Index 2), which corresponds to high-curvature or linear edge features.

3. **Compositional Geometry (Purple Dashed Box):** Perhaps the most compelling evidence comes from `Table` and `Chair`. Unlike simple primitives, these objects are composite: they consist of flat surfaces (tabletops/seats) and thin legs. Remarkably, the heatmap shows that these classes activate *both* the Planar Subspace (#5) and the Linear/Edge Subspace (#3) simultaneously. This demonstrates that MIND has learned a dictionary of geometric primitives, where complex semantic objects are represented as a composition of fundamental error modes (e.g., $Table \approx Plane + Edge$).

These observations strongly support the narrative that $K = 16$ serves as a sufficient capacity for a "Geometric Error Dictionary." By disentangling these primitives, MIND can effectively transfer noise correction patterns from abundant classes (e.g., `Wall`) to rare classes with shared geometry (e.g., `Board`), explaining its superior zero-shot and few-shot robustness.

