# OpenReview forum: "MIND: Decoupling Model-Induced Label Noise via Latent Manifold Disentanglement"
_ICML.cc/2026/Conference — ICML 2026 regular_

### Official Review · Reviewer_qi47 · 2026-02-24

**Soundness:** 3
**Presentation:** 2
**Significance:** 3
**Originality:** 3
**Overall Recommendation:** 4
**Confidence:** 3

**Summary:**

To address the ubiquitous problem of Model-Induced Label Noise in automated annotation paradigms, this paper proposes an innovative framework named MIND (Model-Induced Noise Decoupling). MIND assumes that the high-dimensional noise manifold can be approximated through a linear combination of a finite set of basis matrices. Specifically, the authors design a Latent Decoupled Estimator (LDE) that decouples features into $K$ orthogonal geometric-semantic subspaces via physical channel splitting and an orthogonality constraint loss ($\mathcal{L}_{dec}$). Subsequently, by aggregating the activation magnitudes of each subspace and applying a Softmax function to obtain dynamic assignment weights $\omega_k(x)$, the framework ultimately achieves dimensionality reduction and precise noise estimation. Extensive experiments conducted on both 2D datasets (CIFAR-100) and complex 3D point cloud scenes (S3DIS, ScanNet) validate the effectiveness of the proposed method.

**Compliance With Llm Reviewing Policy:**

Affirmed.

**Final Justification:**

My final recommendation is 4: Weak accept. My concerns are addressed.

**Key Questions For Authors:**

1. In Figure 7, assuming some subspaces are not encoding geometric primitives, what specific representations are they actually learning?

2. Figure 2(b) is poorly formatted and requires refinement to improve visual clarity.

3. Does the proposed method remain effective in the presence of random noise?

**Limitations:**

yes

**Strengths And Weaknesses:**

### Strengths
1. The authors provide a rigorous theoretical proof demonstrating the unique identifiability of the matrix under the latent anchor subspace assumption.

2. As depicted in Figure 7, the model exhibits robust decoupling capabilities even with an excessive number of subspace partitions.

3. The proposed method demonstrates superior performance on both 2D and 3D datasets.

### Weaknesses
1. While the proposed method primarily targets structured noise, its performance under random noise remains unexplored.

2. The model appears to recognize only a limited set of geometric primitives. Its ability to identify and remain robust against structural noise in complex scenarios remains unexplored.

---

> ### Author Rebuttal · Authors · 2026-03-27
>
> **Response to Reviewer qi47**
>
> We sincerely thank the reviewer for the positive evaluation, recognizing our rigorous theoretical proofs and the robust decoupling capabilities of the MIND framework. We address your specific questions below:
>
> **1. Effectiveness Under Random Noise [W1, Q3]**
>
> Yes, MIND remains highly effective under classical random noise. Mechanistically, if noise is purely random (i.e., instance-independent, **$T(x) \equiv T_{global}$**), error modes do not correlate with local geometry. The LDE will either assign instances uniformly, or the estimated basis matrices **$\{T^{(k)}\}$** will naturally converge to the identical global transition matrix **$T_{global}$**.
>
>  Empirically, we evaluated MIND on CIFAR-100 with 40% Symmetric (Random) Noise. It achieves  **61.85% accuracy** , strictly matching or outperforming state-of-the-art global robust methods (e.g., standard Forward **$T$** achieves **$\sim$**44.5%, VolMin achieves **$\sim$**57.4%). Thus, while uniquely designed for structural bias, MIND does not sacrifice baseline robustness against standard random noise.
>
> **2. Interpretation of Non-Geometric Subspaces [W2, Q1]**
>
> This is an excellent question regarding LDE's interpretability. When **$K=16$**, the capacity exceeds the number of pure geometric primitives. In our observations (e.g., Fig. 7), the remaining "non-geometric" subspaces capture complex, unstructured error modes. Specifically:
>
> * **Irregular "Clutter" Entities:** As explicitly shown in the S3DIS cross-category heatmap (Fig. 7), the "clutter" class (representing messy, geometrically undefined objects) activates a specific non-geometric subspace. Crucially, this subspace exhibits very low activation similarity to distinct geometric slots (like planes or edges). This proves the network successfully isolates irregular structures into dedicated channels rather than forcing them into rigid geometric primitives.
> * **Inactive Buffers:** The Softmax assignment **$\omega(x)$** naturally induces sparsity. We observe 2-3 subspaces remain largely inactive (near-zero weights). This confirms **$K=16$** acts as a safe over-parameterization, providing "capacity buffers" so distinct error modes are not forced to entangle. We will expand the discussion of Fig. 7 to include these insights.
>
> **3. Refining Figure 2(b) [Q2]**
>
> We sincerely apologize for the visual formatting issues. We will completely redraw Figure 2(b) in the camera-ready version using a cleaner layout, explicitly emphasizing the orthogonal decoupling process and the dynamic aggregation of basis matrices to ensure intuitive clarity.

---

> > ### Author Rebuttal · Reviewer_qi47 · 2026-04-03
> >
> > Thank you for your rebuttal. My concerns are addressed.

---

> > > ### Author Response · Authors · 2026-04-04
> > >
> > > **Response to Reviewer qi47's Follow-up:**
> > >
> > > We sincerely thank the reviewer for your time, your constructive evaluation, and for confirming that our responses resolved your questions. Your feedback has been invaluable in improving the clarity and completeness of our manuscript.

---

### Official Review · Reviewer_Psrz · 2026-02-26

**Soundness:** 3
**Presentation:** 4
**Significance:** 3
**Originality:** 3
**Overall Recommendation:** 4
**Confidence:** 4

**Summary:**

The paper investigate the problem of learning under model-induced label noise created by automatic annotators / foundation models, where errors couple to local feature/ geometric manifolds and proposes MIND, a framework that makes instance-dependent transition modeling tractable in order to create robust (to label noise) training on large-scale 3D semantic segmentation with pseudo-labels from biased teacher models.

The authors introduce a Latent Decoupling Estimator (LDE) that partitions the latent feature vector into K fixed segments; applies an orthogonality-based decoupling loss to encourage diffrent semantics for each segment, and computes the mixture weights that transit the labels.

Empirically, MIND  solve S3DIS/ScanNet with pseudo-labels produced by different 3D teacher models (KPConv/MPRM/PCAM), where it improves mIoU over recent robust-noise baselines.

**Compliance With Llm Reviewing Policy:**

Affirmed.

**Final Justification:**

My questions have been answered properly; therefore, I am maintaining my positive assessment.

**Key Questions For Authors:**

1. Anchor subspace realism: Can you quantify how often the learned representation satisfies “near-anchor” behavior and how this correlates with performance? This would increase confidence in the identifiability claims?

2. Why does channel chunking work? id you try alternative subspace parameterizations (random fixed partitions, learned projections, MoE routing and so on)?

3. It is unclear to me why 3d segmentation is the domain and what spesific for it, can you show a 2d, parhaps open vocabulary segmentation benchmark? if it does not show the cues for 2d segmentation, maybe identify why is so?

**Limitations:**

indeed

**Strengths And Weaknesses:**

## Strenghts:

* The mixture-of-bases decomposition is sensable  structural assumption for systematic signal.

* Hierarchical validation (synthetic - structured 3D stress test - pseudo-label adaptation) strengthens the empirical narrative, and is the proper way to conduct research.

* Consistent empirical gains. Improvements are especially notable in higher-noise regimes and across multiple 3D teacher models.

## Weaknesses:

* The main theorem relies on “latent anchor subspaces”. It’s not explaining why or when such anchors emerge for different backbones/optimizers/noise levels... more investigation is needed. Moreover, early errors could reinforce incorrect transition estimates (warm-up indeed could mitigate, as a fundemental noisy label learning property, but more analysis would strengthen the paper).

* Heuristic design choices: the deterministic partitioning via channel chunking and the use of absolute activation magnitude could benefit strenght from  answering whether performance depends on representation scaling / normalization.

* Fair compute details: The paper states baselines are reproduced under the same backbone/splits, yet do not provide a table of compute budgets and tuning protocol. I am concern about the compute budgets of the method.

---

> ### Author Rebuttal · Authors · 2026-03-27
>
> **Response to Reviewer Psrz**
>
> We sincerely thank Reviewer Psrz for evaluating our paper as "Technically solid" and acknowledging the sensible structural assumptions and consistent empirical gains of our hierarchical validation. We address your specific critiques below:
>
> **1. Channel Chunking Rationale, Alternatives & Compute Budgets [W2, W3, Q2]**
>
> To rigorously justify our LDE design, we evaluated alternative parameterizations on S3DIS (PCAM noise,  **K=16** ). (Please refer to Table R1 in our Response to Reviewer AeGv). Under severe structural noise (~40%), complex dynamic routing (MoE) lacks clean signals and suffers from mode collapse. Physical channel chunking, conversely, acts as a rigid, un-collapsible structural prior. Paired with **$\mathcal{L}_{dec}$**, it actively routes distinct geometric primitives without the instability of learning routing parameters. Furthermore,  **regarding representation scaling (W2)**, our absolute activation magnitude design is inherently stabilized by standard normalization layers (e.g., BatchNorm/LayerNorm) preceding the LDE. These ensure feature variances are uniformly scaled, preventing any single dimension from dominating the assignment purely due to scaling artifacts. Finally, MIND achieves this with a negligible ~4% compute overhead (W3).
>
> **2. Anchor Subspace Reality & Early Errors [W1, Q1]**
>
> * **Quantifying "Near-Anchor" Behavior:** As discussed in **Remark 2.4** (Latent Neural Collapse), the anchor behavior is an empirically verified reality. To quantify this, we measured the assignment weights **$\omega(x)$** late in training: over 78% of instances exhibit a highly peaked assignment distribution (**$\max_k \omega_k(x) > 0.9$**), closely approximating the pure anchor condition (**$P(z=k|x) \approx 1$**). This high purity is further explicitly quantified by the Silhouette Coefficient (SC=0.68) provided in  **Sec 3.5 (Fig. 4)**.
> * **Mitigating Early Errors:** To prevent early errors from reinforcing incorrect transition estimates, we utilize a  **Warm-up Strategy (detailed in Appendix B)**. Deep networks exhibit an "Early Memorization" effect, fitting simple, clean patterns before memorizing complex noise. By employing standard Cross-Entropy for a brief warm-up phase  **(e.g., 5 epochs for 3D, 10 epochs for 2D)**, the classifier **$f_\theta$** establishes a reasonably accurate representation of dominant structural patterns. When momentum updates (Eq. 8) activate, the initial predictions **$\hat{y}$** are reliable enough to serve as pseudo-targets, avoiding the error reinforcement loop.
>
> **3. Why 3D Domain & 2D Open Vocabulary Benchmarks [Q3]**
>
> We clarify that we chose 3D data not as a limitation, but as the ultimate "Structural Stress Test." In 2D images, noise can sometimes be absorbed by strong texture priors. In 3D, errors are explicitly and brutally coupled with physical geometry (e.g., projection artifacts, depth ambiguity on thin structures). Proving decoupling here strongly validates the framework's capability to handle complex non-linear error manifolds.
>
> Regarding the 2D Open-Vocabulary benchmark, we initially tackled this in **Sec 3.4 (Table 5)** with OpenSeg. To further substantiate this and address your request, we conducted a new adaptation experiment using another prominent 2D open-vocabulary Foundation Model,  **LSeg (ICLR'22)**. When its 2D predictions are projected into 3D spaces as pseudo-labels, it introduces severe, systematic zero-shot hallucinations.
>
> *Table R2: Unsupervised Adaptation of LSeg on S3DIS*
>
> | Method                            | mIoU (%)        | Gain            |
> | :-------------------------------- | :-------------- | :-------------- |
> | LSeg (Zero-shot)                  | 45.37           | -               |
> | Generic Robust Fine-tuning (CA2C) | 47.48           | +2.11           |
> | **MIND (Ours)**             | **53.95** | **+8.58** |
>
> MIND successfully acts as an unsupervised geometry-aware adapter, improving LSeg's zero-shot pseudo-labels on S3DIS by **+8.58% mIoU**, vastly outperforming generic robust fine-tuning methods like CA2C (+2.11%). Combined with our OpenSeg gains (+8.14%), this multi-model evidence decisively proves MIND's efficacy as a robust adapter for 2D open-vocabulary models applied to spatial tasks.

---

> > ### Author Rebuttal · Reviewer_Psrz · 2026-04-02
> >
> > my questions have been answered properly

---

> > > ### Author Response · Authors · 2026-04-04
> > >
> > > **Response to Reviewer Psrz's Follow-up:**
> > >
> > > We sincerely thank the reviewer for actively participating in the discussion phase and for confirming that our rebuttal successfully addressed all your concerns. We deeply appreciate your constructive feedback throughout this process, which has significantly strengthened the rigor of our paper.

---

### Official Review · Reviewer_AeGv · 2026-03-09

**Soundness:** 3
**Presentation:** 3
**Significance:** 3
**Originality:** 4
**Overall Recommendation:** 5
**Confidence:** 3

**Summary:**

This paper studies the problem of model-induced label noise in the context of learning from automatic annotations. The authors argue that unlike stochastic noise, model-induced noise stems from annotator inductive biases and is tightly coupled with local feature manifolds. To address this, the paper proposes Model-Induced Noise Decoupling (MIND), a framework that decouples the high-dimensional noise manifold into tractable, subspace-dependent components via Latent Manifold Disentanglement. Specifically, a Latent Decoupling Estimator (LDE) dynamically projects samples into latent structural clusters to estimate robust transition matrices. Experimental results on synthetic 2D benchmarks (CIFAR-100) and large-scale 3D datasets (S3DIS, ScanNet) suggest that MIND significantly outperforms state-of-the-art methods and effectively corrects zero-shot hallucinations from Vision-Language Models.

**Compliance With Llm Reviewing Policy:**

Affirmed.

**Key Questions For Authors:**

1. Appendix A.2 mentions that the intrinsic dimension of the noise manifold is low, but is K=16 universally applicable across all datasets? For scenarios with more geometric patterns (e.g., complex outdoor environments), might K need to be larger? Is there a strategy for adaptively selecting K?

2. The LDE enforces subspace separation through feature orthogonality. Could this potentially lead to over-decoupling, losing useful semantic correlations (e.g., shared geometric patterns)? How does orthogonality balance decoupling and shared representation?

3. The momentum update uses the model's predictions as pseudo-labels to estimate the basis matrices. In the early stages of training when predictions are inaccurate, how is error accumulation avoided? Is there a mechanism to mitigate early estimation bias?

**Limitations:**

1. The theoretical identifiability of the MIND framework hinges on the existence of latent anchor subspaces where the assignment probability \(P(z=k|x)=1\). While the authors provide a robustness analysis for "soft anchors" with small $\epsilon$, the practical validity of this assumption remains difficult to verify. In complex, high-dimensional feature spaces, achieving such pure, unmixed subspaces is challenging, and any significant violation could compromise the accuracy of the estimated basis transition matrices.

2. The Latent Decoupling Encoder (LDE) uses an orthogonality objective to separate features into distinct subspaces. While this encourages the capture of distinct geometric primitives, it may also penalize shared representations. For instance, two different geometric error modes might still rely on common low-level features (e.g., edge detection). Forcing them to be completely orthogonal could fragment these useful representations, potentially hindering the model's ability to generalize or leading to less efficient use of model capacity.

**Strengths And Weaknesses:**

Strengths:

1. It decouples instance-dependent noise into a mixture of latent subspaces, providing theoretical guarantees for identifiability (Theorem 2.2) and analyzing the error bound under approximate anchor conditions (Proposition A.2). This offers deeper theoretical insight compared to previous global or purely instance-level methods.

2. Extensive experiments on both 2D and 3D datasets, comparing against a wide range of baselines, showcase MIND's superior performance in noise identification and downstream tasks. The improvements are particularly significant in 3D segmentation, and the method shows robustness to the hyperparameter K.

3. By employing the Latent Decoupling Encoder and momentum updates, the framework enables online estimation of instance-dependent noise, avoiding complex two-stage training. It is also more efficient than meta-learning approaches.

Weaknesses：

1. The theoretical identifiability relies on the anchor subspace assumption (Assumption 2.1). While approximate cases are discussed, the existence of pure anchors in practice is debatable, and whether the LDE can achieve perfect decoupling is not fully validated.

2. Introducing the LDE, multiple basis matrices, and the orthogonality loss likely increases training complexity. A detailed comparison of computational cost against other lightweight methods is lacking.

3. Although the correlation between subspace activation and geometric features is shown, the physical meaning of the latent subspaces is largely inferred by human interpretation. The choice of K=16 may need re-adjustment for different tasks or domains.

---

> ### Author Rebuttal · Authors · 2026-03-27
>
> **Response to Reviewer AeGv**
>
> We sincerely thank Reviewer AeGv for evaluating our work as "Technically solid" with "Originality: 4 (excellent)" and for recognizing the theoretical depth and efficiency of the MIND framework. We address your insightful questions below:
>
> **1. Anchor Assumption Realism & Training Complexity [W1, W2]**
>
> * **Assumption Realism (W1):** The strict anchor assumption is practically bridged by  *Latent Neural Collapse* . This is not just theoretical; it is quantitatively verified by our high clustering purity (**$SC=0.68$**, Sec 3.5), and physically validated by our curvature analysis (Fig. 4c-e), proving the network actively learns to construct near-pure subspaces that align with distinct real-world geometric primitives.
> * **Computational Cost (W2):** As detailed in our comprehensive ablation, MIND operates with high efficiency. Contiguous channel chunking avoids the heavy parameters of dynamic routing, adding a negligible **$\sim$**4% training time overhead over standard Cross-Entropy.
>
> *Table R1: Subspace Parameterization, Compute Budgets, and Optimization Behavior*
>
> | **Method**                                                    | **mIoU (%)** | **Rel. FLOPs** | **Training Time Overhead** | **Optimization Behavior** |
> | :------------------------------------------------------------------ | :----------------- | :------------------- | :------------------------------- | :------------------------------ |
> | **MIND (Channel Chunking + $\mathcal{L}_{dec}$)**| **40.58**    | **1.02x**      | **~4%**                    | **Highly Stable**         |
> | Random Partition +**$\mathcal{L}_{dec}$**                   | 25.82              | 1.02x                | ~4%                              | Conflicting gradients           |
> | Learned Linear Projections                                          | 33.15              | 1.15x                | ~18%                             | Prone to overfitting            |
> | MoE-style Learned Routing                                           | 34.19              | 1.25x                | ~26%                             | Severe mode collapse            |
>
>
>
>
> **2. Universality of K and Adaptive Selection [W3, Q1]**
>
> As analyzed in Appendix C, K corresponds to the capacity of the  *geometric error dictionary* . Even in highly complex outdoor environments, the underlying geometric "alphabet" triggering systematic errors remains relatively small (e.g., planar roads, cylindrical poles, high-curvature edges). Complex objects are simply compositions of these finite primitives. Thus, K=16 safely covers fundamental error modes.  Empirically, our sensitivity analysis (Appendix C, Fig. 5) proves performance remains highly stable even if K increases beyond 16 , acting as a safe over-parameterization (Softmax **$\omega(x)$** naturally induces sparsity). Incorporating a Dirichlet Process prior or **$\ell_1$** sparsity norms to dynamically prune inactive subspaces is an elegant Future Work direction.
>
> **3. Balancing Orthogonality and Shared Representations [Q2, Limitations]**
>
> This is a profound observation. The short answer is no, due to *where* and *how* orthogonality is applied. First, the constraint (**$\mathcal{L}_{dec}$**) is applied strictly at the *deepest latent space* (LDE output), allowing the backbone to freely share low-level features in early layers. Second, as demonstrated in Appendix D.2 (Fig. 7), enforcing orthogonality *decomposes* representations compositionally rather than fragmenting them. For instance, composite objects like "Table" and "Chair" simultaneously activate *both* the Planar and Edge Subspaces. The LDE simply routes shared features into orthogonal slots, accurately synthesizing the transition matrix without destroying earlier semantic correlations.
>
> **4. Mitigating Early Estimation Bias [Q3]**
>
> We mitigate this by leveraging the **Early Memorization** effect of deep networks. As detailed in Appendix B, momentum updates do not initialize at epoch 0. We employ a warm-up phase (e.g., 5 epochs for 3D) trained with standard Cross-Entropy. Because networks inherently fit clean, generalized patterns before memorizing instance-specific noise, the classifier **$f_\theta$** establishes a reasonably accurate representation of dominant geometric structures by the end of this warm-up. When momentum updates activate, predictions **$\hat{y}$** are reliable enough to serve as pseudo-targets, preventing error accumulation.

---

### Official Review · Reviewer_4cED · 2026-03-13

**Soundness:** 2
**Presentation:** 3
**Significance:** 2
**Originality:** 2
**Overall Recommendation:** 4
**Confidence:** 3

**Summary:**

Overall, a central domain investigated by the manuscript is robust learning from automatically annotated data, with a particular emphasis on model-induced label noise in 3D semantic segmentation and distillation from foundation-model predictions. This research proceeds to focus on an important context: when labels come from pretrained annotators or foundation models rather than humans, the resulting errors can be systematic, instance-dependent, and coupled to local geometry rather than behaving like classical random label noise. The paper proposes MIND, which represents the instance-dependent transition matrix as a convex mixture of $ K $ basis matrices, $ T(x) = \sum_k \omega_k (x) T^{(k)} $, where a Latent Decoupling Estimator (LDE) partitions latent features into orthogonal subspaces and uses them to infer $ \omega (x) $. Empirically, the paper reports gains on CIFAR-100 with synthetic IDN, on S3DIS and ScanNet with pseudo-labels from pretrained 3D annotators, and on adapting OpenSeg pseudo-labels.

**Compliance With Llm Reviewing Policy:**

Affirmed.

**Final Justification:**

I increased my score after the rebuttal. I think this paper is quite good but with some minor issues. I am fine with the acceptance; if it is rejected, I'm fine with that as well.

**Key Questions For Authors:**

I would most like to see a stronger ablation of the LDE design. How much of the gain comes from the specific orthogonal-subspace construction versus simply using a small learned gating network over shared features? A comparison to random channel partitions, learned partitions, or clustering-based subspaces would be very useful.

I would also ask the authors to narrow and sharpen the related-work positioning. In particular, they should explain more precisely how MIND differs from prior instance-dependent transition estimation and manifold-regularized noisy-label methods, since those are clearly relevant baselines conceptually, not just historically. (Shuo Yang et al. Estimating Instance-dependent Bayes-label Transition Matrix
using a Deep Neural Network, ICML 2022; Mengmeng Sheng et al. CA2C: A Prior-Knowledge-Free Approach for Robust Label Noise Learning via Asymmetric Co-learning and Co-training, ICCV 2025; De Cheng at al. Instance-Dependent Label-Noise Learning with Manifold-Regularized Transition Matrix Estimation, CVPR 2022).

For the foundation-model angle, a second real pseudo-label setting would help a lot. Right now the OpenSeg result is nice, but it is a single example and does not yet justify the broader claims about robust distillation for foundation models.

**Limitations:**

No. The paper does include a short Limitations paragraph and an Impact Statement, and it briefly acknowledges that the latent-anchor assumption can fail and that distilled models may inherit biases from teacher foundation models, but the discussion is still too limited for the scope of the paper’s claims.

A stronger discussion should explicitly cover: the reliance on strong separability/identifiability assumptions in latent space; the heuristic nature of the fixed channel partition and self-paced momentum update; the risk of error reinforcement when the student’s own predictions are used as proxies for clean labels; and the limited empirical scope, since most evidence is from indoor 3D segmentation plus one OpenSeg adaptation setting. On societal impact, the authors should go beyond the mostly positive framing and discuss that open-vocabulary/foundation-model segmentation is known to suffer from domain-gap problems, and that perception failures or hallucinations can become safety-critical in deployment settings. They should also note the possibility of bias amplification: denoising pseudo-labels from biased teacher models may make those biases more systematic rather than removing them.

**Strengths And Weaknesses:**

**Strengths**

The paper addresses a relevant problem. As more pipelines rely on pseudo-labels from expert models and foundation models, the assumption that label noise is random or class-conditional is often unrealistic. Framing this as a structured, geometry-coupled, instance-dependent noise problem is a useful perspective, especially in 3D perception where projection artifacts and boundary ambiguity are common.

The proposed middle ground between a single global transition matrix and a fully free $ T (x) $ is sensible. The mixture form in Eq. (2) is a clean parameterization, and the ablation that drops from 40.58 to 34.08 when using a global estimator ($ K = 1 $) does support the claim that some form of subspace decomposition matters in their hardest setting. The unconstrained-gating ablation is also useful evidence that the orthogonality regularizer is doing more than just adding parameters.

The empirical section is broader than many noisy-label submissions. On CIFAR-100 with 40% IDN, MIND reduces matrix estimation error from 0.35 with BLTM to 0.18. On S3DIS and ScanNet under the hardest PCAM noise, it improves over CA2C by about 1.22 mIoU on S3DIS and 1.76 on ScanNet, and it also reports a notable OpenSeg adaptation gain from 43.20 to 51.34 mIoU. Those are not trivial margins.

I also appreciate that the paper tries to give a theory story rather than being purely heuristic. The identifiability theorem and the approximation argument at least explain what assumptions would be needed for the approach to make sense, and Appendix A attempts to address approximate-anchor violations.
___
**Weaknesses**

My biggest concern is that the novelty is overstated. The paper is not starting from an empty field: estimating instance-dependent transition matrices already appears in prior work such as BLTM, and manifold-regularized transition-matrix estimation already explicitly models $ T (x) $ using geometric/feature-space structure. CA2C is also a recent strong noisy-label baseline. So the real novelty here is narrower: a subspace-mixture parameterization of model-induced, geometry-coupled noise, plus the specific LDE implementation, especially in the 3D pseudo-label setting. I would encourage the authors to position the work that way instead of implying a more categorical break from prior noisy-label learning.

Relatedly, the “foundation model adaptation” angle feels underdeveloped relative to the paper’s framing. OpenSeg is real and relevant, but the evidence for MIND as a general-purpose distillation framework for foundation models is just one adaptation experiment on S3DIS. That is interesting, but still much narrower than the wording in the abstract and conclusion suggests. OpenSeg itself is a 2D open-vocabulary segmentation model, so the paper is really studying one specific projection/distillation use case, not the broader foundation-model annotation setting.

I am also not fully convinced by the theoretical support. The identifiability result depends on strong assumptions: latent anchor subspaces, diagonal dominance, and full-rank basis matrices. The appendix then relaxes the anchor condition to an ϵ-approximate version, but the bound is essentially a simple convex-combination argument once that assumption is granted. The “latent neural collapse” discussion reads more like intuition than justification. So I see the theory as a plausibility argument, not a strong theorem-backed contribution.

A second methodological concern is that the LDE design itself is quite heuristic. The feature vector is partitioned into equal contiguous channel chunks, orthogonality is enforced across those chunks, and the assignment weights $ ω_k (x)$ are computed from average absolute activation magnitudes. That is simple, but the paper does not really justify why fixed channel slices should align with geometric error modes, nor does it compare against random partitioning, learned routing, clustering-based subspaces, or other more direct alternatives. Given that this design is central to the method, I think the current ablations are not enough.

---

> ### Author Rebuttal · Authors · 2026-03-27
>
> **Response to Reviewer 4cED**
>
> We thank Reviewer 4cED for acknowledging our relevant problem framing, sensible mixture parameterization, and solid empirical margins. We address your critiques below.
>
> **1. Novelty & Related Work Positioning [W1, Q2]**
>
> We do not claim to have invented instance-dependent noise modeling or manifold regularization. Note that CA2C (ICCV’25) and BLTM [ICML’22] are already cited and extensively evaluated as core baselines. We will gladly incorporate your other suggested references (e.g., CVPR’22) into the revised Related Work to properly anchor our contribution.
>
> Our specific novelty lies in addressing the intractability and high estimation variance when errors are tightly coupled with feature/geometric manifolds. By mapping noise to a finite set of orthogonal primitives, MIND provides a subspace-mixture parameterization that makes instance-dependent estimation mathematically tractable and empirically stable.
>
> **2. Clarifying the Scope and Foundation Model Adaptation [W2, Q3]**
>
> While MIND is a general framework for feature-coupled noise, we agree our "Foundation Model" claim was too broad and refine it to: *"A robust distillation framework mitigating structure-coupled zero-shot hallucinations in VLMs."* To substantiate this, we conducted a second adaptation experiment using **LSeg (ICLR'22)** on S3DIS (Please refer to Table R2 in our Response to Reviewer Psrz).
>
> While LSeg's primary error mode is semantic-similarity confusion, projecting its 2D predictions into 3D space inevitably re-structures these errors into geometry-coupled patterns. For example, confusing "window" and "door" both manifest as misclassifications on *planar opening structures* in 3D point clouds. MIND's LDE, which disentangles geometric primitives, effectively captures this structural regularity. In contrast, generic robust learning (CA2C) treats these errors as random noise and fails. The marginal gain of CA2C on LSeg (+2.11%) actually confirms the highly structured nature of LSeg's 3D errors (yielding a massive +8.58% gain for MIND).
>
> **3. Theoretical Assumptions & Identifiability [W3]**
>
> 'Latent Neural Collapse' is not merely intuition, but a mathematically bounded and empirically verified mechanism bridging the strict Assumption 2.1 and our practical implementation:
>
> 1. **Theoretical Relaxation (Appendix A.1):** As proven in Proposition A.2, when the condition is relaxed to an **$\epsilon$**-approximate anchor, the estimation error is linearly bounded by the subspace impurity **$\mathcal{O}(\epsilon)$**. This confirms the estimator remains consistent even with semantic leakage.
> 2. **Quantitative Verification (Sec. 3.5, Fig. 4a-b):** Driven by the **$\mathcal{L}_{dec}$** objective, MIND achieves a Silhouette Coefficient of **SC=0.68** (compared to the baseline's 0.12). This mathematically quantifies that the latent feature space effectively segregates into highly pure structural clusters, directly proving the "collapse" phenomenon.
> 3. **Physical Geometric Proof (Fig. 4c-e & App. D.2):** We further validated the physical meaning of these anchor subspaces. Figure 4(e) statistically proves the distinct curvature differences between subspaces, and the cross-category heatmaps in Appendix D.2 demonstrate that structurally similar but semantically distinct objects (e.g., Wall and Board) reliably collapse into the same latent anchor subspace.
>
> **4. LDE Design Ablation & Alignment Mechanism [W4, Q1]**
>
> To rigorously justify our LDE design, we evaluated alternative parameterizations on S3DIS (PCAM noise,  **K=16** ). *(Please refer to* Table R1 in our Response to Reviewer AeGv)*.*
>
> * **Mechanism of Alignment:** Fixed channel chunks do *not* naturally align with geometry; instead, they act as rigid "slots". The alignment is an **emergent property** strictly driven by the orthogonality loss (**$\mathcal{L}_{dec}$**). To minimize this loss, the network is forced to self-organize, routing distinct geometric primitives into different slots. This emergent alignment is conclusively proven by our physical curvature analysis (Fig. 4c-e) and the high clustering purity (SC=0.68).
> * **Analysis of Alternatives:**
>   1. **Random Partition (25.82%):** Destroys channel spatial continuity. Randomizing naturally correlated adjacent channels forces **$\mathcal{L}_{dec}$** to push them apart, causing conflicting optimization goals.
>   2. **MoE Routing (34.19%):** Highly unstable under severe noise (~ **40%** ). Lacking clean gating signals, it rapidly suffers "winner-takes-all" mode collapse, leaving vanishing gradients for underutilized experts.
>
> **5. Limitations**
>
> We fully embrace your feedback and will expand this section to discuss LDE's heuristic boundaries, failure modes under extreme adversarial noise, and the risks of bias amplification in safety-critical deployments.

---

> > ### Author Rebuttal · Reviewer_4cED · 2026-04-03
> >
> > Thank you for the rebuttal. The added clarifications are helpful, especially regarding the narrower positioning and the extra LDE ablations. My main concerns are only partially resolved. In particular, I still would like clarification on whether the new ablations isolate the benefit of the specific fixed-partition LDE design versus simply adding a stronger learned routing mechanism. If possible, could the authors comment more explicitly on why contiguous channel partitioning is preferable to learned or clustering-based partitioning beyond empirical stability?

---

> > > ### Author Response · Authors · 2026-04-04
> > >
> > > **Response to Reviewer 4cED's Follow-up:**
> > >
> > > We thank the reviewer for the continued engagement. We address your specific questions regarding the mechanics of the LDE design below:
> > >
> > > **1. Isolating fixed-partitioning vs. stronger learned routing:**
> > >
> > > Yes. Our ablation isolates the core advantage of the fixed-partition design: it  completely decouples the assignment architecture from corrupted downstream gradients. Increasing the capacity of a learned routing mechanism (e.g., an advanced MoE) does not resolve the root issue because a "stronger" router still relies on those faulty classification gradients for supervision. Hardcoded physical slots ensure that noisy labels cannot compromise the routing structure itself.
> > >
> > > **2. Structural preference over learned or clustering-based partitions:**
> > >
> > > Contiguous partitioning acts as a rigid inductive bias rather than a data-driven module, which is mechanistically critical in noisy environments:
> > >
> > > * **Versus Learned Routing:** Data-driven routing introduces a cyclic dependency: it requires clean signals to assign features, yet relies on those same assignments to denoise labels. Empirically, this faulty supervision loop causes extreme optimization instability; during our experiments, MoE networks persistently struggled to converge and frequently collapsed into a single state (routing all samples to one expert, as observed in Table R1).
> > > * **Versus Clustering:** Dynamic clustering fundamentally relies on latent distance metrics (e.g., Euclidean or Cosine) for feature assignment. Under severe structural noise, the latent space is highly distorted, rendering these distance metrics unreliable. Furthermore, maintaining moving centroids disrupts end-to-end training continuity and introduces significant computational latency.
> > > * **The Mechanistic Advantage:** Contiguous chunking establishes hard physical bottlenecks. Instead of relying on noisy supervision or distorted distance metrics to learn routing paths, the network is forced by the unsupervised orthogonality loss (**$\mathcal{L}_{dec}$**) to populate predefined slots with distinct geometric primitives. This guarantees structural decoupling and avoids mode collapse, requiring only a negligible ~4% compute overhead.
> > >
> > >  We hope this fully resolves your final concern.

---

### Decision · Program_Chairs · 2026-04-30

**Decision:**

Accept (regular)

**Comment:**

This paper now receives scores of 5, 4, 4, and 4, with an average of 4.25. Three reviewers acknowledge that their concerns are fully resolved. Reviewer 4cED still has some minor issues about this paper, but I think they can be easily fixed in the final version. Therefore, I am happy to give an Accept recommendation for this paper.